# Multilevel Analysis of Individual and Contextual Factors Associated with Polio Non-Vaccination in Africa: Further Analyses to Enhance Policy and Opportunity to Save More Lives

**DOI:** 10.3390/vaccines9070683

**Published:** 2021-06-22

**Authors:** Olalekan A. Uthman, Duduzile Ndwandwe, Muhammed M. B. Uthman, Sanni Yaya, Charles S. Wiysonge

**Affiliations:** 1Warwick Centre for Global Health, Division of Health Sciences, University of Warwick Medical School, Coventry CV4 7AL, UK; 2Department of Global Health, Division of Epidemiology and Biostatistics, Faculty of Health Sciences, Stellenbosch University, Francie van Zijl Drive, Tygerberg, Cape Town 7505, South Africa; Charles.Wiysonge@mrc.ac.za; 3Cochrane South Africa, South African Medical Research Council, Francie van Zijl Drive, Parow Valley, Cape Town 7501, South Africa; duduzile.ndwandwe@mrc.ac.za; 4Department of Epidemiology & Community Health, College of Health Sciences, University of Ilorin, Ilorin 240211, Nigeria; uthmanmb@yahoo.com; 5School of International Development and Global Studies, University of Ottawa, Ottawa, ON K1N 6N5, Canada; hsanniya@uottawa.ca; 6The George Institute for Global Health, Imperial College London, London SW7 2AZ, UK; 7Division of Epidemiology and Biostatistics, School of Public Health and Family Medicine, Faculty of Health Sciences, University of Cape Town, Anzio Road, Observatory, Cape Town 7925, South Africa

**Keywords:** polio, vaccination, neighbourhood, multilevel analysis, Africa

## Abstract

**Background:** Africa was certified polio-free in 2020 and to maintain the polio-free status, African countries need to attain and maintain optimal routine polio vaccination coverage. One indicator for optimal polio vaccination coverage is the prevalence of children who have received no polio vaccination through routine services. The objective of the study was to examine the individual-, neighbourhood-, and country-level factors associated with non-vaccination against polio in Africa. **Methods:** We applied multivariable multilevel logistic regression analyses on recent demographic and health survey data collected from 2010 onwards in Africa. We identified 64,867 children aged 12–23 months (Level 1) nested within 16,283 neighbourhoods (Level 2) from 32 countries (Level 3). **Results:** The prevalence of non-vaccination for polio ranged from 2.19% in Egypt to 32.74% in Guinea. We found the following factors to be independent predictors of the increased odds of non-vaccination for polio: being a male child, born to mother with no formal education, living in poorer households; being from a polygamous family, living in neighbourhoods with high maternal illiteracy, high unemployment rate, and low access to media. **Conclusions:** We found that both individual and contextual factors are associated with non-vaccination for Polio.

## 1. Introduction 

It is undeniable that the use of vaccines has prevented more premature deaths, permanent disability, and suffering, in all regions of the world, than any other medical discovery or intervention [1,2]. Immunization has saved over 20 million lives in the last two decades. More than 100 million infants are immunised each year, saving 2–3 million lives annually. Poliomyelitis (polio) mainly affects children under five years of age. One in two hundred infections leads to irreversible paralysis. 

Wild poliovirus cases have decreased by over 99% since 1988, from an estimated 350,000 cases in more than 125 endemic countries to 179 reported cases in two countries in 2019 [3]. Polio remains endemic in two countries—Afghanistan and Pakistan [4]. Until poliovirus transmission is interrupted in these countries, all countries remain at risk of importation of polio, especially vulnerable countries with weak public health and immunisation services and travel or trade links to endemic countries. Of the three strains of wild poliovirus (type 1, type 2, and type 3), wild poliovirus type 2 was eradicated in 1999, and no case of wild poliovirus type 3 has been found since the last reported case in Nigeria on 25 August 2020 [5]. Nigeria brought the world one major step closer to ending a disease that has paralyzed millions of people worldwide. 

As the world grapples with a variety of programme and policy challenges related to childhood immunisation, we believe that a comprehensive and relevant evidence base that would equip states to take informed actions is a critical component in changing the status quo. A lot of research has been done on the individual compositional factors that are correlated with childhood immunisation coverage. Indeed, we found no published research that looked at the social factors that contributed to childhood polio non-vaccination. This omission is significant despite the important role of communities in shaping parenting habits, as they influence individual opportunities and link residents to a variety of risks and resources throughout their lives. The aim of this study was to examine individual-level compositional factors as well as contextual factors measured at the community and country levels that are associated with polio non-vaccination. 

## 2. Methods

### 2.1. Study Design

Data for this cross-sectional study were obtained from Demographic and Health Surveys (DHS), which are nationally representative household surveys conducted in Africa. This study used data from 32 recent DHS surveys conducted from 2010 in African countries available as of March 2021. Demographic and Health Surveys (DHS) are nationally representative household surveys that provide data for a wide range of monitoring and impact evaluation indicators in the areas of population, health, and nutrition.

### 2.2. Sampling Technique

The DHS employs a stratified, multistage sampling approach, with homes serving as the sampling unit [6]. All women and men who match the eligibility criteria are interviewed in each sample household. Weights are calculated to account for differential selection probabilities as well as non-response because the surveys are not self-weighting. The results of the survey, when weighted, represent the entire target population. A household questionnaire, a women’s questionnaire, and, in most countries, a men’s questionnaire are all included in the DHS surveys.

DHS surveys collect primary data using four different types of model questionnaires. A household questionnaire is used to gather information about the characteristics of the household’s dwelling unit as well as the characteristics of regular residents and visitors. It is also used to identify household members who are eligible for an individual interview. individual woman’s or man’s questionnaire is then used to interview eligible respondents. The biomarker questionnaire is used to gather biomarker information from children, women, and men. The woman’s questionnaire collects data on the following topics: background characteristics, reproductive behaviour and intentions, contraception, antenatal, delivery, and postnatal care, breastfeeding and nutrition, children’s health, status of women, HIV and other sexually transmitted infections, husband’s background, and other topics: questions examine behaviour related to environmental health, the use of tobacco, and health insurance. 

### 2.3. Data Collection

To achieve comparable data across countries, all DHS questionnaires were implemented with similar interviewer training, supervision, and implementation protocols. Procedures for collecting data have been published elsewhere [6]. In a nutshell, data were gathered by visiting households and conducting face-to-face interviews to obtain information on maternal and child health indicators, among other things.

### 2.4. Ethical Consideration

This study is based on an analysis of existing survey data with all identifier information removed. The surveys were approved by the Ethics Committee of the ICF at Rockville, MD, in the USA and by the corresponding National Ethics Committee in the Ministries of Health from each country. All study participants gave informed consent before participation, and all information was collected confidentially.

### 2.5. Outcome Variable

Children who have received no vaccines through routine vaccination services are referred to as zero-dose children, which we refer to in this study as polio non-vaccination. We chose the term polio non-vaccination to avoid confusion with the birth dose of polio, which is often referred to in African countries as “polio zero dose”. Non-vaccinated child for polio was described as a binary variable that takes the value 1 if a child aged 12–23 months has not received any of the four routine doses of oral polio vaccine (polio 0 at birth, polio 1 at 6 weeks, polio 2 at 10 weeks, and polio 3 at 14 weeks) and 0 otherwise.

### 2.6. Explanatory Variables

#### 2.6.1. Individual Level Factors

We included the following individual level factors: child’s age (in months), child sex (male or female), high birth order (less than 24 months), number of under-five children, polygamous family, mother’s age (completed years) wealth index (poorer, middle, or richer), mother’s and father’s education (no education, primary, secondary, or higher), employment status (working or not working), has health insurance, media access (access to radio, television, or newspaper), and maternal health-seeking behaviours (prenatal visits, tetanus injection during pregnancy, medical assistance at delivery, knowledge of oral rehydration solution (ORS), and possession of a health card for the child). The DHS did not collect any direct data on household’s income and spending. As a proxy for socioeconomic status, we used the DHS wealth index. The methods used to calculate the DHS wealth index have previously been defined [7,8]. In brief, an index of economic status was created for each household utilizing principal components analysis based on the following household variables: number of rooms per home, ownership of a vehicle, motorcycle, bicycle, fridge, television, and telephone, as well as any type of heating system. The tertiles of the DHS wealth index (poor, middle, and rich) were estimated and used in the subsequent modelling.

#### 2.6.2. Neighbourhood-Level Factors

We used the word “neighbourhood” to describe a grouping of people who live in the same geographical area [9]. Within the DHS data, neighbourhoods were defined based on the presence of a common primary sample unit [10].

The models contained the following neighbourhood-level factors: *Neighbourhood poverty:* percentage of households below 20% of wealth index*Illiteracy rate*: percentage of women with no formal education in the community*Unemployment rate*: percentage of women not working in the community*Media access*: percentage of households with access to television, radio, or newspaper*Average household size:* mean number of people in each community*Female-headed households*: percentage of households headed by women in an area.*Residential mobility:* proportion of households occupied by persons who had moved from another dwelling in the previous 5 years [11,12,13]*Place of residence:* urban or rural, as administratively defined by each country*Ethnic diversity*—an index of ethnic diversity was created using a formula that captures both the number of different groups in an area and the relative representation of each group [14]:
$$\mathrm{Ethnic}\;\mathrm{diversity}\;\mathrm{index}=1-\overset{n}{\underset{i=1}{\sum }}{\left[\begin{array}{c} \underline{x_{i}}\\ y \end{array}\right]}^{2}$$
where:$x_{i}$ = population of ethnic group *i* of the area, *y* = total population of the area, *n* = number of ethnic groups in the area

Scores can range from 0 to approximately 1. For clarity of interpretation, each diversity index is multiplied by 100; the larger the index, the greater diversity there is in the area. If an area’s entire population belongs to one ethnic group, then an area has zero diversity. An area’s diversity index increases to 100 when the population is evenly divided into ethnic groups.

#### 2.6.3. Country-Level Factors

Data at the country level were gathered from reports released by the United Nations Development Program [15]. We included the intensity of deprivation at the national level, which is the average percentage of deprivation faced by people living in multidimensional poverty, and this was categorized into two (low and high). We categorised community- and country-level variables into two categories (low and high) to allow for non-linear effects and provide more readily interpretable results in the policy arena. Median values served as the reference group for comparison.

### 2.7. Control Variable

The year the DHS was conducted was included as a partial control for a period trend to control for effects of unknown factors that may have been introduced due to different timing of surveys across countries.

### 2.8. Statistical Analyses

#### 2.8.1. Descriptive Analyses

In the descriptive statistics, respondents’ distribution by key variables was expressed as percentages. 

#### 2.8.2. Modelling Approaches

We used multivariable logistic multilevel regression models to analyse the association between individual compositional and contextual factors associated with polio non-vaccination. We specified a 3-level model for binary response reporting children that did not receive any polio vaccine (at level 1), in a neighbourhood (at level 2) living in a country (at level 3). 

Five models were built. The first model, a null model with no explanatory variables, was used to decompose the amount of variance between country and neighbourhood levels. The second model only included individual-level factors, the third model only included neighbourhood-level factors, and the fourth model only included country-level factors. Finally, the fifth model accounted for person-, neighbourhood-, and country-level factors all at the same time (full model).

#### 2.8.3. Fixed Effects (Measures of Association)

Fixed-effect findings (measures of association) were recorded as odds ratios (ORs) with 95 percent credible intervals (CrIs). 

#### 2.8.4. Random Effects (Measures of Variation)

The possible contextual effects were measured by the intra-class correlation (ICC) and median odds ratio (MOR). We measured similarity between respondents in the same neighbourhood and within the same country using ICC. The ICC represents the percentage of the total variance in the probability of non-vaccination for polio that is related to the neighbourhood- and country-level, i.e., measure of clustering of odds of non-vaccination for polio in the same neighbourhood and country. The ICC was calculated by the linear threshold (latent variable method) [16]. Following the ideas of Larsen et al. on neighbourhood effects [17], we reported the random effects in terms of odds. The MOR measures the second or third level (neighbourhood or country) variance as odds ratio and estimates the probability of non-vaccination for polio attributed to neighbourhood and country context. MOR equal to one indicates no neighbourhood or country variance. Conversely, the higher the MOR, the more important are the contextual effects for understanding the probability of non-vaccination for polio. 

#### 2.8.5. Model Fit and Specifications

We examined the multicollinearity among explanatory variables. The multilevel models were fitted using the MLwinN programme, version 2.31 [18,19]. We used the Bayesian Deviance Information Criterion to measure how well different models fitted the data [20].

## 3. Results

### 3.1. Sample Characteristics

The DHS surveys were conducted between 2010 and 2018. As shown, in Figure 1 and Figure 2, the prevalence of non-vaccination for polio varied across the countries. It ranged from 9.9% to 32.7% in Central Africa; from 3.6% to 29.8% in Western Africa; from 4.4% to 20.8% in Eastern Africa; and from 7.0% to 12.7% in Southern Africa. Table 1 presents the descriptive statistics for the final pooled sample. For this analysis, we analysed information on 64,867 children aged 12–23 months (Level 1) nested within 16,283 neighbourhoods (Level 2) from 32 countries (Level 3) in African countries. Half of the children were male. Almost four in ten mothers had no formal education, and one in three were from poorer households. In addition, one third of the women were not working at the time of the survey (30.1%). Only about one in ten had health insurance (11%). About half of the respondents were living in high poverty rates. 

### 3.2. Measures of Associations (Fixed Effects)

The results of different models are shown in Table 2. In the fully adjusted model controlling for the effects of individual-, neighbourhood- and societal-level factors; male children were 9% more likely to be non-vaccinated for polio compared with female children (odds ratio [OR] = 1.09; 95% credible interval [CrI] 1.03 to 1.16). Children of women with no education (OR = 1.43, 95% CrI 1.29 to 1.59) and from poorest household (OR = 1.30, 95% CrI 1.19 to 1.42) were 43% and 30% more likely to be non-vaccinated for polio, respectively. Children of women not currently working (OR = 1.18, 95% CrI 1.10 to 1.28), no health insurance (OR = 1.22, 95% CrI 1.03 to 1.44), and that do not seek healthcare (OR = 7.52, 95% CrI 6.88 to 8.21) were 18%, 22%, and 652% more likely to be non-vaccinated for polio, respectively. 

Children living in neighbourhood with high maternal illiteracy rate (OR = 1.26, 95% CrI 1.17 to 1.36), high unemployment rate (OR = 1.08, 95% CrI 1.00 to 1.16), and high non-access to media (OR = 1.12, 95% CI 1.04 to 1.21) were 26%, 8%, and 12% more likely to be non-vaccinated for polio, respectively. 

### 3.3. Measures of Variations (Random Effects)

As shown in Table 2, in Model 1 (unconditional model), there was a significant variation odd of non-vaccinated for polio across the countries ($\sigma^{2}=$ 0.73, 95% CrI 0.43–1.24) and across the neighbourhoods ($\sigma^{2}=$ 1.25, 95% CrI 1.15–1.37). According to the intra-country and intra-neighbourhood correlation coefficient, 13.9% and 37.6% of the variance in odds of non-vaccinated for polio could be attributed to the country- and neighbourhood-level factors, respectively. The full model accounted for 42.5% and 44.9% of the variance in the in odds of non-vaccinated for polio across the countries and neighbourhood. Results from the median odds ratio (MOR) also confirmed evidence of neighbourhood and societal contextual phenomena shaping odds of non-vaccinated for polio. From the full model (Model 5), it was estimated that if a child moved to another country or another neighbourhood with a higher probability of non-vaccinated for polio, there would be 1.86-fold (95% CrI 1.61–2.26) and 2.21-fold (95% CrI 2.07–2.32) increase in their odds of being non-vaccinated for polio. 

## 4. Discussion

To our knowledge, the current study is the first multilevel examination of non-vaccination for polio in Africa countries using national representative data and a very large number of children. We found that male children, born to mothers with no formal education, mothers not currently working, mothers with no health insurance, mothers that do not seek healthcare for their children, mothers from poorer households, from neighbourhoods with high maternal illiteracy, high unemployment rate, and low access to media were more likely to be non-vaccinated for polio. Our findings are not unique. Previous findings from the United States show that under vaccination was more likely in male than female children [21]. Similarly, studies conducted in African countries indicated that children born to mothers with no formal education, children who resided in the poorest households [22], children whose households reported distance to the nearest health facility as a big problem, children whose delivery occurred in a non-health facility, and children who had no health card were more likely to be non- and under-vaccinated [23,24]. More importantly, the findings uncover new evidence by demonstrating that neighbourhood and country-level factors influence non-vaccination for polio above and beyond individual level factors. 

We found evidence of geographical clustering in non-vaccination for polio. About 60% and 81% of the variation in non-vaccination for polio, is conditioned by differences between neighbourhoods and countries, respectively. Suppose a respondent moved to another neighbourhood or another country with a higher probability of non-vaccination for polio. In that case, their odds of non-vaccination for polio may increase by about 3.0- and 4.7-fold, respectively. It is instinctual that children from the same neighbourhood may be more similar to each other in relation to their non-vaccination for polio than to others from other neighbourhoods [25], i.e., a contextual phenomenon which expresses itself as clustering of individual non-vaccination for polio within a neighbourhood [25]. This phenomenon of clustering behaviour was also reported by Khowaja and colleagues, who found that polio vaccine refusals clustered in certain people from specific income levels and cultural backgrounds in Karachi, Pakistan [26]. The authors found that low-income people from an ethnic group known as Pashtuns were less likely to participate in polio supplementary immunisation activities than low-income non-Pashtuns. Reasons commonly cited among Pashtuns for refusing vaccination included fear of sterility, lack of faith in the polio vaccine, scepticism about the vaccination programme, and fear that the vaccine might contain religiously forbidden ingredients [26]. On these grounds, we might conclude that there is some evidence for a possible neighbourhood and country contextual phenomenon shaping a common children odds of non-vaccination for polio. These findings underscore the need to implement public health prevention strategies not only at the high-risk individual level but also in high-risk neighbourhoods. 

Our findings should be interpreted with the following caveats in mind. To begin with, the cross-sectional nature of the data restricts our ability to draw causal inferences from the observed association. Second, we did not assess the amount of time participants spent in their communities. As a result, we were unable to assess if the associations of community features with polio non-vaccination were due to cumulative effects. Finally, one significant drawback is that DHS surveys do not collect information on household income [7,27]. 

Notwithstanding these limitations, the research has important strengths. It is a massive population-based study with national coverage in 56 countries and a strong response rate. Furthermore, variables in DHS were operationalized in the same way, allowing numerical values to be comparable across countries. The Bayesian approach we used has the added benefit of producing a much more accurate estimation with better properties and yielding unbiased estimates [28,29,30]. 

## 5. Conclusions

In summary, we found that non-vaccinated children for polio tended to be male, have mothers with no formal education, mothers who are not working, mothers with no health insurance, mothers that do not seek healthcare for their children, mothers from poorer households, and mothers from neighbourhoods with high maternal illiteracy, high unemployment rate, and low access to media. Our findings highlight the need for strategies in routine childhood vaccination to consider contextual barriers and clustering behaviour resulting in children not receiving the life-saving intervention to protect them against polio.

## Figures and Tables

**Figure 1 vaccines-09-00683-f001:**
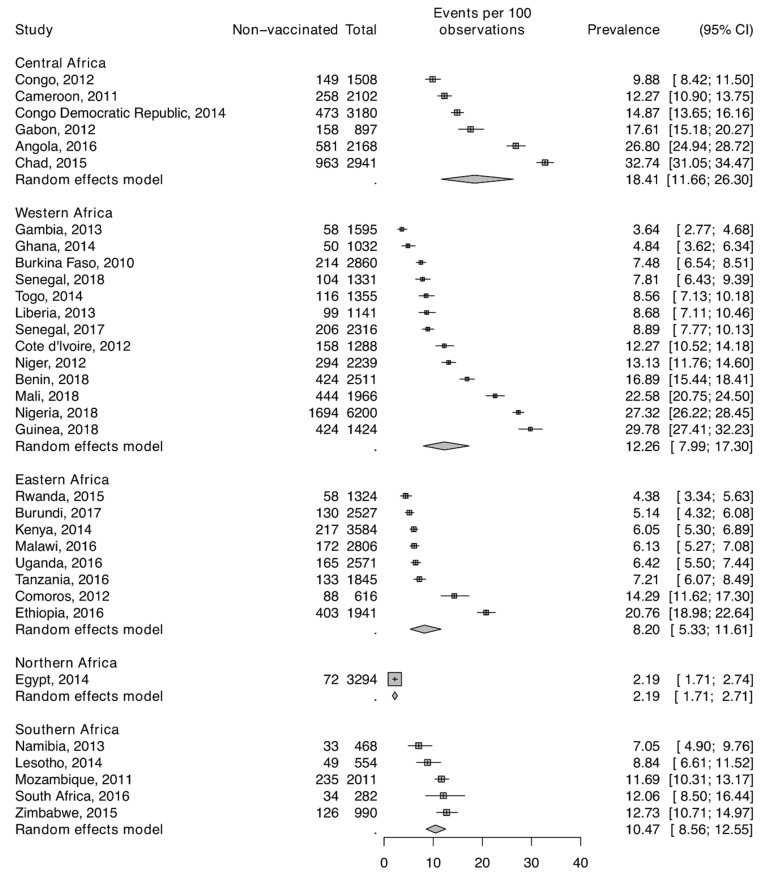
Prevalence in non-vaccination for polio across African regions.

**Figure 2 vaccines-09-00683-f002:**
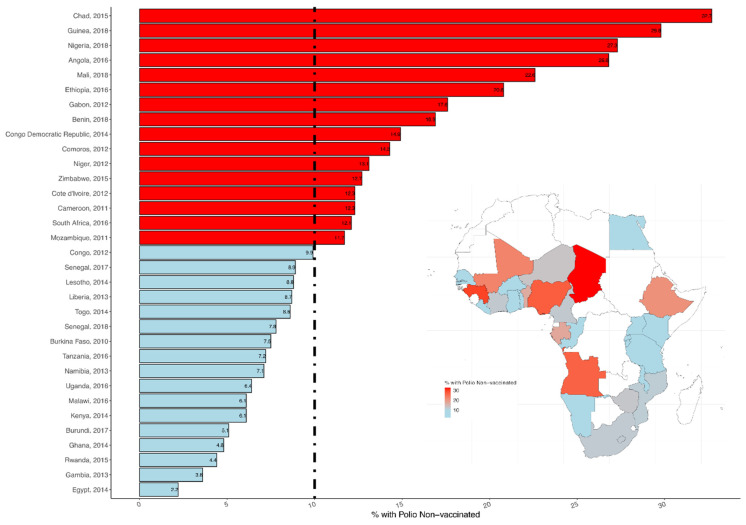
Prevalence in non-vaccination for polio across African countries.

**Table 1 vaccines-09-00683-t001:** Summary of pooled sample characteristics of the Demographic and Health Surveys data in Africa, 2010–2018.

Variable	Number (%)
Male	31,918 (49.2)
Child’s age, months median (IQR)	17.0 (6.0)
Mother’s age, years median (IQR)	28.0 (10.0)
Child’s birth order	
1	12,271 (18.9)
2	12,370 (19.1)
3	10,808 (16.7)
4	8705 (13.4)
5 or more	20,713 (32.0)
Under-5 children	
1	19,853 (30.6)
2	26,759 (41.2)
3	9672 (14.9)
4	3354 (5.2)
5 or more	3312 (5.1)
Mother education	
No education	22,014 (41.6)
Primary	20,652 (31.8)
Secondary or higher	17,195 (26.5)
Father education	
No education	22,193 (35.3)
Primary	17,000 (27.0)
Secondary or higher	23,736 (37.7)
Wealth index	
Poorer	21,633 (33.3)
Middle	21,601 (33.3)
Richer	21,633 (33.3)
Currently working	44,845 (69.1)
Has health insurance	3536 (5.5)
Polygamous family	16,712 (25.8)
Maternal health seeking index	
0	4155 (6.4)
1	2784 (4.3)
2	7870 (12.1)
3	21,122 (32.6)
4	28,936 (44.6)
Media access	
0	21,831 (33.7)
1	22,047 (34.0)
2	15,756 (24.3)
3	5233 (8.1)
Neighbourhood-level factors	
Urban	19,272 (29.7)
High vs. low community diversity	23,721 (36.6)
High vs. low neighbourhood poverty rate	32,414 (50.0)
High vs. low female head	30,489 (47.0)
High vs. low residential instability	15,406 (23.8)
High vs. low illiteracy rate	31,352 (48.3)
High vs. low unemployment rate	31,202 (48.1)
High vs. low no neighbourhood media access	
Average household size	
Country-level factors	
HDI	4941 (7.6)

**Table 2 vaccines-09-00683-t002:** Individual compositional and contextual factors associated with non-vaccination for polio identified by multivariable multilevel logistic regression models, Demographic and Health Surveys data, 2010–2018.

	Model 1	Model 2 OR (95% CrI)	Model 3 OR (95% CrI)	Model 4 OR (95% CrI)	Model 5 OR (95% CrI)
Survey year		**1.04 (1.04, 1.04)**	**1.09 (1.09, 1.09)**		**1.06 (1.06, 1.06)**
Individual-level					
Male (vs) female		**1.09 (1.03, 1.16)**			**1.09 (1.03, 1.16)**
Child’s age		**1.01 (1.00, 1.02)**			1.01 (1.00, 1.01)
Child’s birth order		1.01 (0.97, 1.04)			1.00 (0.97, 1.03)
Mother’s age		**0.99 (0.99, 1.00)**			0.99 (0.99, 1.00)
Mother education					
No education		**1.16 (1.04, 1.30)**			**1.43 (1.29, 1.59)**
Primary		**1.15 (1.03, 1.28)**			**1.31 (1.20, 1.144)**
Secondary or higher		1 (reference)			1 (reference)
Father education					
No education		**1.18 (1.07, 1.30)**			**1.30 (1.19, 1.42)**
Primary		**1.12 (1.03, 1.23)**			**1.17 (1.08, 1.27)**
Secondary or higher		1 (reference)			1 (reference)
Wealth index					
Poorer		0.98 (0.89, 1.07)			**1.30 (1.19, 1.45)**
Middle		1.00 (0.91, 1.09)			**1.14 (1.06, 1.25)**
Richer		1 (reference)			1 (reference)
Not currently working		**1.22 (1.13, 1.30)**			**1.18 (1.10, 1.28)**
No health insurance		**1.21 (1.12, 1.31)**			**1.22 (1.03, 1.44)**
Polygamous family		**1.42 (1.32, 1.52)**			**1.41 (1.32, 1.50)**
No Maternal health seeking		**7.85 (7.20 (8.56)**			**7.52 (6.88, 8.21)**
No media access		**1.12 (1.05, 1.19)**			**1.07 (1.00, 1.15)**
Neighbourhood-level					
Urban vs. rural			**1.20 (1.10, 1.31)**		1.00 (0.90, 1.09)
High vs. low community diversity			1.02 (0.95, 1.09)		1.01 (0.93, 1.09)
Neighbourhood poverty rate			**1.28 (1.18, 1.38)**		0.96 (0.88, 1.06)
Female headed households			**0.93 (0.87, 0.99)**		1.01 (0.94, 1.09)
Residential instability rate			**0.91 (0.83, 1.00)**		1.04 (0.94, 1.14)
Illiteracy rate			**1.85 (1.72, 1.99)**		**1.26 (1.17, 1.36)**
Unemployment rate			**1.20 (1.12, 1.29)**		**1.08 (1.00, 1.16)**
Neighbourhood media access			**1.36 (1.27, 1.47)**		**1.12 (1.04, 1.21)**
Average household size			1.00 (0.98, 1.01)		1.00 (0.99, 1.01)
Societal-level					
Human development index				0.54 (0.26, 1.15)	0.94 (0.49, 1.81)
Random effects					
Country-level					
Variance (95% CrI)	0.73 (0.43, 1.24)	0.29 (0.17, 0.49)	0.70 (0.42, 1.16)	0.67 (0.39, 1.12)	0.42 (0.25, 0.73)
VPC	13.9 (8.8, 21.0)	6.5 (4.0, 10.4)	14.0 (9.0, 21.1)	12.8 (8.1, 19.4)	9.5 (6.1, 15.2)
MOR	2.26 (1.87, 2.89)	1.66 (1.47, 1.95)	2.22 (1.85, 2.79)	2.18 (1.81, 2.75)	1.86 (1.61, 2.26)
Explained variation (%)	reference	61.0 (60.6, 61.4)	4.7 (3.5, 6.5)	9.0 (9.3, 9.6)	42.5 (41.1, 41.9)
Neighbourhood-level					
Variance (95% CrI)	1.25 (1.15, 1.37)	0.82 (0.72, 0.91)	0.98 (0.90, 1.05)	1.25 (1.16, 1.36)	0.69 (0.58, 0.78)
VPC	37.6 (32.4, 44.2)	25.0 (21.2, 29.8)	33.7 (28.5, 40.2)	36.8 (32.0, 43.0)	25.2 (20.1, 31.4)
MOR	2.91 (2.78, 3.05)	2.37 (2.25, 2.48)	2.57 (2.47, 2.66)	2.91 (2.79, 3.04)	2.21 (2.07, 2.32)
Explained variation (%)	reference	34.8 (33.7, 37.3)	21.8 (21.6, 23.4)	0.0 (−1.0, 0.7)	44.8 (43.1, 49.4)
Model fit statistics					
DIC	44,124	35,937	43,585	44,128	35,909
Sample size					
Country-level	32	32	32	32	32
Neighbourhood-level	16,283	15,929	16,283	16,283	15,929
Individual-level	64,867	62,923	64,867	64,867	62,923

Model 1—empty null model, baseline model without any explanatory variables (unconditional model); Model 2—adjusted for only individual-level factors; Model 3—adjusted for only neighbourhood-level factors; Model 4—adjusted for only country-level factors; Model 5—adjusted for individual-, neighbourhood-, and country-level factors (full model); OR—odds ratio, CrI—credible interval, MOR—median odds ratio, VPC—variance partition coefficient, and DIC—Bayesian deviance information criteria.

## Data Availability

The data supporting this article are available at: http://dhsprogram.com/data/available-datasets.cfm (accessed on 27 March 2021).

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
