# Peer review of "Multilevel Analysis of Individual and Contextual Factors Associated with Polio Non-Vaccination in Africa: Further Analyses to Enhance Policy and Opportunity to Save More Lives"

_vaccines, 2021, doi:10.3390/vaccines9070683_

Round 1

Reviewer 1 Report

This article is interesting and already very successful, I have some suggestions for the authors

Author Response

c

Thank you for the pertinent comments which have helped to improve the quality of this manuscript.

Line 70: Year is missing

Thanks this been added “on 25 August 2020”

Line 120: This part would deserve to be a little more developed, on how the data are collected, on how the household, women and men questionnaire are developed.

More information on the DHS Survey would also be appreciated

Thanks, we have now re-written the description of the DHS data extensively under the following sub-headings: study design, sampling technique, data collection and ethical consideration.

Line 219: there is a layout problem here

Thanks, it must be due to the software the online system used in converting the document to PDF. This has been corrected.

Discussion:

Discussion parts needs to be a little more developed, I find the explanation / interpretation of the results too quick. There is little comparison with other studies.

The limitation part of the study should be better identified and a little more developed.

A conclusion paragraph is missing.

Thanks, we have now strengthened the discussion section. We have added the following sentences:

“This phenomenon of clustering behaviour was reported by Khowaja and colleagues (2012) that polio vaccine refusals clustered in low-income Pashtun (9.8%) and high-income families of any ethnic background (46.4%). Low-income Pashtuns were more likely not to have participated in polio supplementary immunization activities than low-income non-Pashtuns (odds ratio, OR: 7.1; 95% confidence interval, CI: 3.47-14.5). Reasons commonly cited among Pashtuns for refusing vaccination included fear of sterility; lack of faith in the polio vaccine; scepticism about the vaccination programme, and fear that the vaccine might contain religiously forbidden ingredients(31).”

“Our findings highlight the need for strategies in vaccination campaigns to consider contextual barriers and clustering behaviour resulting in children not receiving the life-saving intervention to protect them against polio.’

Reviewer 2 Report

Comments to the Author:

Dear Editor,

Thank you for the opportunity to revise the manuscript entitled: “Children Who Received No Polio Vaccine in Africa: Further Analyses to Enhance Policy and Opportunity to Save More Lives” submitted by Uthman OA et al. The manuscript aims to highlight the social factors that contribute to childhood polio non-vaccination. This type of information, which lacks in published literature, is really relevant and can help to address public health prevention strategies.

Section comments:

The paper is interesting, covering an important topic. The statistical methods are appropriate. Tables and figures are very explanatory and supportive in the interpretation of the results. However, there are some aspects that Authors should address before publication. Please, see below.

The manuscript does not indicate the study’s design in the title or abstract. In the title, more specifically, it should be addressed that the study wants to examine the social factors associated with non-vaccination.

The abstract is probably too long, it is 400+ words, with the suggested length being approximately 200 words. The results subheading is too long, it should summarize the article’s main findings.

The keywords are absent.

Main criticisms:

The introduction is too long and with not strictly useful information. Authors should consider condensing the introduction. If appropriate, I would suggest using some of this information in the discussion.

In the methods, I would suggest adding some extra information on the DHS. As for instance, eligibility criteria (Line 115), and how the weight was calculated. Moreover, the Authors spoke about three different types of questionnaires. Can you add some fore insights? Can you also add info on ethical approval?

Discussion is too short and without comparison with previous evidence. Please, add some consideration on similarities or differences between your results and data already available in the literature. It could be interesting also to add differences and similarities not only with other African studies but also with other studies conducted in other low-income countries. Discussion and conclusions should be more focused on public health implications. What are the added values of this work? How policymakers can use these data to implement preventive policies? How vaccination campaigns can benefit from these results?

Minor and more formal criticisms:

Lines 63-64: This is redundant information.

Line 70: There is a missing year.

Lines 76-78: This is redundant information.

Lines 85-88: This information should require a reference regarding the Immunization Strategic Plan.

Line 125: Layout error, “Explanatory…” should be in line 126.

Line 132: “polygamous family” is redundant information.

Line 148: The layout is wrong, the number one should be before “neighbourhood poverty”, and all should be in the lower line.

Line 163: “and” should be removed.

Line 172: typos “which is the the average…”

Lines 219-221: This paragraph is bold and the results heading should start a new line.

Line 223: The sentence is not clear.

Line 242-244: Between the sentence “In the fully adjusted model controlling for the effects of individual-, neighbourhood- and societal-level factors.” and “Male children were 9% more likely to be non-vaccinated for polio compared with female children (odds ratio [OR] = 1.09; 95% credible interval [CrI] 1.03 to 1.16).” there is a dot that I think it is not needed.

Line 273: typos, “-fold”.  

Author Response

Comments to the Author:

Dear Editor,

Thank you for the opportunity to revise the manuscript entitled: “Children Who Received No Polio Vaccine in Africa: Further Analyses to Enhance Policy and Opportunity to Save More Lives” submitted by Uthman OA et al. The manuscript aims to highlight the social factors that contribute to childhood polio non-vaccination. This type of information, which lacks in published literature, is really relevant and can help to address public health prevention strategies.

Thank you for the pertinent comments which have helped to improve the quality of this manuscript.

Section comments:

The paper is interesting, covering an important topic. The statistical methods are appropriate. Tables and figures are very explanatory and supportive in the interpretation of the results. However, there are some aspects that Authors should address before publication. Please, see below.

Thanks, all the comments we received on this study have been taken into account in improving the quality of the article.

The manuscript does not indicate the study’s design in the title or abstract. In the title, more specifically, it should be addressed that the study wants to examine the social factors associated with non-vaccination.

Thanks, we have now change the title of the manuscript to include the study design and factors “Multilevel analysis of individual and context factors associated with polio non-vaccination in Africa: Further Analyses to Enhance Policy and Opportunity to Save More Lives”.

The abstract is probably too long, it is 400+ words, with the suggested length being approximately 200 words. The results subheading is too long, it should summarize the article’s main findings.

Thanks, we have now edited the abstract down to 198 words.

The keywords are absent.

Now added the following Keywords: Polio, vaccination, neighbourhood, multilevel analysis, Africa

Main criticisms:

The introduction is too long and with not strictly useful information. Authors should consider condensing the introduction. If appropriate, I would suggest using some of this information in the discussion.

Thanks, the introduction has been revised and shortened to 329 words.

In the methods, I would suggest adding some extra information on the DHS. As for instance, eligibility criteria (Line 115), and how the weight was calculated. Moreover, the Authors spoke about three different types of questionnaires. Can you add some fore insights? Can you also add info on ethical approval?

Thanks, we have now re-written the description of the DHS data: study design, sampling technique, data collection and ethical consideration.

“Study design

Data for this cross-sectional study were obtained from Demographic and Health Surveys (DHS), which are nationally representative household surveys conducted in Africa. This study used data from 32 recent DHS surveys conducted from 2010 in African countries available as of March 2021. Demographic and Health Surveys (DHS) are nationally representative household surveys that provide data for a wide range of monitoring and impact evaluation indicators in the areas of population, health, and nutrition.

Sampling technique

The DHS employs a stratified, multi-stage sampling approach, with homes serving as the sampling unit  (6). All women and men who match the eligibility criteria are interviewed in each sample household. Weights are calculated to account for differential selection probabilities as well as non-response because the surveys are not self-weighting. The results of the survey, when weighted, represent the entire target population. A household questionnaire, a women's questionnaire, and, in most countries, a men's questionnaire are all included in the DHS surveys.

DHS surveys collect primary data using four different types of Model Questionnaires. A Household Questionnaire is used to gather information about the characteristics of the household's dwelling unit as well as the characteristics of regular residents and visitors. It is also used to identify household members who are eligible for an individual interview. Individual Woman's or Man's Questionnaire is then used to interview eligible respondents. The Biomarker Questionnaire is used to gather biomarker information from children, women, and men. The Woman's Questionnaire collects data on the following topics: background characteristics, reproductive behaviour and intentions, contraception, antenatal, delivery, and postnatal care, breastfeeding and nutrition, children's health, Status of women, HIV and other sexually transmitted infections, husband's background, and other topics: questions examine behaviour related to environmental health, the use of tobacco, and health insurance.

Data collection

To achieve comparable data across countries, all DHS questionnaires were implemented with similar interviewer training, supervision, and implementation protocols. Procedures for collecting data have been published elsewhere  (6). In a nutshell, data were gathered by visiting households and conducting face-to-face interviews to obtain information on maternal and child health indicators, among other things.

Ethical consideration

This study is based on an analysis of existing survey data with all identifier information removed. The surveys were approved by the Ethics Committee of the ICF at Rockville, MD in the USA and by the corresponding National Ethics Committee in the Ministries of Health from each country. All study participants gave informed consent before participation and all information was collected confidentially.”

Discussion is too short and without comparison with previous evidence. Please, add some consideration on similarities or differences between your results and data already available in the literature. It could be interesting also to add differences and similarities not only with other African studies but also with other studies conducted in other low-income countries. Discussion and conclusions should be more focused on public health implications. What are the added values of this work? How policymakers can use these data to implement preventive policies? How vaccination campaigns can benefit from these results?

Thanks, we have now strengthened the discussion section. We have added the additional sentences.

Minor and more formal criticisms:

Lines 63-64: This is redundant information.

Thanks, this has been deleted

Line 70: There is a missing year.

We have now added this.

Lines 76-78: This is redundant information.

Deleted, thanks.

Lines 85-88: This information should require a reference regarding the Immunization Strategic Plan.

The sentence is deleted in this revised draft

Line 125: Layout error, “Explanatory…” should be in line 126.

Thanks, it must be due to the software the online system used in converting the document to PDF. This has been corrected.

Line 132: “polygamous family” is redundant information.

Deleted, thanks.

Line 148: The layout is wrong, the number one should be before “neighbourhood poverty”, and all should be in the lower line.

Thanks, it must be due to the software the online system used in converting the document to PDF. This has been corrected.

Line 163: “and” should be removed.

Deleted, thanks.

Line 172: typos “which is the the average…”

Deleted, thanks.

Lines 219-221: This paragraph is bold and the results heading should start a new line.

Thanks, it must be due to the software the online system used in converting the document to PDF. This has been corrected.

Line 223: The sentence is not clear.

This sentence has been re-written “As shown, in figures 1 and 2, the prevalence of non-vaccination for polio varied across the countries’

Line 242-244: Between the sentence “In the fully adjusted model controlling for the effects of individual-, neighbourhood- and societal-level factors.” and “Male children were 9% more likely to be non-vaccinated for polio compared with female children (odds ratio [OR] = 1.09; 95% credible interval [CrI] 1.03 to 1.16).” there is a dot that I think it is not needed.

Thanks for pointing that out. The extra dot has been deleted and the two sentences now separated by a comma “In the fully adjusted model controlling for the effects of individual-, neighbourhood- and societal-level factors, male children were 9% more likely to be non-vaccinated for polio compared with female children (odds ratio [OR] = 1.09; 95% credible interval [CrI] 1.03 to 1.16).”

Line 273: typos, “-fold”.  

This has been rephrased “there would be 1.86-fold (95% CrI 1.61 to 2.26) and 2.21-fold (95% CrI 2.07 to 2.32) increase in their odds of non-vaccinated for polio.”

Round 2

Reviewer 2 Report

Thank you for having considered my suggestions. I think that the general quality of the reporting has been improved.